# Does the Adoption of Digital Payment Improve the Financial Availability of Farmer Households? Evidence from China

**Baozhen Chen [1],\* and Jinzheng Ren [2],\***

1   School of Management, Beijing Union University, Beijing 100101, China
2   College of Economics and Management, China Agricultural University, Beijing 100083, China
\*   Correspondence: gltbaozhen@buu.edu.cn (B.C.); rjzheng@cau.edu.cn (J.R.)

**Abstract:** Digital finance carries the expectation of achieving inclusiveness. The purpose of this paper is to explore how digital finance can improve the financial availability and the extent to which digital finance can improve the financial availability of farmer households. Based on micro-rural survey data in China from 2017 to 2019, employing the Cov-AHP weighting method, this study measured the index of financial availability (IFA) of farmer households in terms of three dimensions: investment, bank loans, and private finance. We analyzed the mechanism of how digital payment adoption affects the IFA of farmer households based on the Long Tail Effect theory of Anderson. Ordinary least squares method and ordered probit model was constructed to empirically test the impact of payment adoption on the IFA of farmer households. The research results show that (1) the IFA of Chinese rural households is still at a low level; (2) while the availability of investment is very low, the availability of bank loans is relatively high; and (3) the adoption of digital payment has a positive impact on improving the IFA of farmer households, including the availability of investments, bank loans, and private finance. The results are robust to model misspecification and reverse causality. The evidence also suggests that the adoption of digital payment mainly affects the financial availability of farmer households through information effects. Therefore, attention should be paid to broadening information channels and promoting the adoption of digital payments to improve financial access for farmer households. This study contributes to the comprehensive understanding of the financial situation of households by constructing a financial availability indicator system from three dimensions. By analyzing the impact of digital payment adoption on farmers' financial availability, this study helps to understand how digital finance can play a positive role in farmer households' financial conditions.

**Keywords:** farmer households; digital payment; financial availability; information effect





## 1. Introduction

A large amount of the extant literature has documented that financial development, especially the development of inclusive finance, has a significant positive impact on economic growth [1–4], reductions in income inequality [4,5], consumption growth [6], and women's empowerment [7]. Financial inclusion offers not only access to credit but also access to an array of financial products and services [8]. Although financial inclusion has become a topical issue, the economic literature on digital financial inclusion is still in its early stages [9]. Previous studies on inclusive finance have carried out calculations and analyses from the macro-level of countries and regions [3–5,7,9]. Rural finance, as the weakest link in inclusive finance, has not received the attention it deserves.

The development of digital finance has been seen as a new and powerful tool to improve rural households' access to finance [10]. In 2016, the "G20 High-Level Principles for Digital Financial Inclusion" was officially passed, becoming the first important guiding principle of the digital economy of global significance. The upgraded version of the "G20 Financial Inclusive Index System" submitted at the same time added new indicators for

digital financial inclusion. The role of digital finance in promoting financial inclusion has been widely recognized worldwide. Digital payment is an important concept and provides strong support to digital finance, and it is also an important channel for the development of digital financial inclusion. According to the "Global Payment Report" released by FIS, among the global e-commerce payment methods in 2020, the share of electronic/mobile wallets reached 44.5%, and these have also been widely adopted in rural areas.

Rural finance is the weakest link in the development of inclusive finance. How to promote and realize the inclusiveness of rural finance is directly related to the overall development of inclusive finance. It is particularly important for China's rural revitalization strategy and the economic development of the entire country. The financial availability of rural households is a concrete manifestation of rural financial inclusiveness at the micro level. The ultimate goal of promoting rural financial inclusiveness is to improve the financial availability of farmer households. As an important means to realize inclusive finance, the extent to which digital payment improves the financial availability of farmer households and how digital payment improves the financial availability of farmer households have become issues worthy of in-depth study and discussion.

On the basis of theoretical analysis, this study empirically tested the impact of digital payment usage on the financial availability of farmer households, including the degree of impact and the impact mechanism. We contribute to the existing literature in the following ways: (1) this study took farmer households as the research object, laying a micro-level foundation for understanding the status quo of rural finance and the development of inclusive finance. (2) Considering that farmer households have equal rights to obtain investment and financial income through asset allocation, the availability of investments is included in the evaluation system. In addition, private finance, as a reservoir for financial services, is a necessary supplement to formal finance and is an important channel that affects farmer households' financial availability. An indicator system was constructed to evaluate the financial availability of rural households from the three dimensions of investment availability, bank loan availability, and the availability of private finance. (3) Based on the long tail effect theory, this study constructed a theoretical analysis framework for the inclusive effects of digital payment and conducted an empirical test of the theoretical framework with survey data from rural China.

## 2. Related Literature and Theoretical Underpinnings

### 2.1. Related Literature

Many organizations and scholars have elaborated on the definition of inclusive finance. The underlying ideas are basically similar, that is, "individuals and businesses have access to useful and affordable financial products and services that meet their needs—transactions, payments, savings, credit and insurance—delivered in a responsible and sustainable way" (Source: The World Bank, https://www.worldbank.org/en/topic/financialinclusion/overview (accessed on 30 June 2022)). In terms of inclusive financial measurement, Beck et al. [11] and Sarma [12] have carried out pioneering work to build a regional inclusive financial indicator system. In their research, the indicator system is mainly reflected in three dimensions: accessibility, availability, and usage of banking services. Financial inclusion can bring many welfare benefits to individuals [13], and scholars have confirmed the positive effects of inclusive finance on income distribution, poverty alleviation, women's empowerment [7], and economic development [14,15].

The problem of rural finance is a worldwide issue and is one of the focuses of inclusive finance. McKinnon [16] and Shaw [17] first demonstrated the problems of financial repression and financial deepening. In summary, rural financial problems stem from the following reasons: a typical inter-separated dual financial structure [18], adverse selection and moral hazard caused by information asymmetry [19], and interest rate control in the case of information asymmetry [20]. Furthermore, farming systems are increasingly facing the unknown, with uncertainty and surprises [21]. Digital finance has the advantages of wide coverage, high speed, and low cost, and can make up for the shortcomings of rural

finance. The concept of digital financial inclusion has also rapidly gained popularity. Many researchers have observed that digital finance can affordably provide financial services for rural and poor groups without access to financial services, and has become an important way to achieve inclusive finance [22].

The integration of information and communication technology (ICT) and finance is conducive to alleviating poverty [23,24], increasing consumption [25,26] and improving farmers' entrepreneurial behavior, which could promote inclusive economic growth [27]. Digital payments refer to any types of payments made using digital instruments, which include mobile payments, mobile wallets, cryptocurrency, and electronic payments [28], which, combined with the ubiquitous mobile phone technology, enable the re-engineering of financial systems, including the use of pre-paid cards, mobile financial apps, mobile banking, etc. [29]. In China, digital payments have been widely used in rural areas, bringing hope for the development of digital financial inclusion. In general, the current research on financial inclusion has mostly been at the national or administrative level [30]. This approach will inevitably weaken the problem of rural financial exclusion. Regarding how to measure the development level of inclusive finance, the current research has mainly focused on farmers' access to basic financial services such as deposit accounts and bank loans. With the development of the economy, farmer households' demand for financial services will become more diversified. In terms of the inclusive effects of digital finance, most scholars have regarded digital finance directly as inclusive finance, focusing on its economic effects, but failing to delve into the influence mechanisms, which limits the applicability of the research results.

### 2.2. Theoretical Underpinnings

According to the Long Tail Effect theory of Anderson [31], the share of tails with small demand but a large number of goods is roughly the same as the share of heads with high demand but a small number of goods. The financial demand of Chinese rural households is relatively small and at the end of the financial market, but it is not zero and scalable, which conforms to the tail characteristics of the long tail effect theory [32].

Digital finance has overturned traditional financial services. Digital finance not only overcomes the physical limitations of traditional finance, but also realizes low-cost real-time information transmission and acquisition. To a certain extent, new financial supplies are generated because of digital technology. The internal mechanism of the adoption of digital payment affecting the investment behavior of farmer households is shown in Figure 1. Farmer households need a certain level of capital, financial knowledge, and venues for investment, which means there are certain thresholds for farmer households to participate in the financial market. According to the status quo of rural financial development, it can be judged that most rural households are below the threshold at this stage in China. We assumed that, before farmers use digital payments, the equilibrium point is $Q_1$ for farmers to invest and manage wealth, and the transaction volume stabilizes at $x_0$. With the introduction of Internet institutions into the financial market, the emergence of monetary funds (such as Yu Ebao, a balance value-added service and current fund management service product by the Ant Group) has increased the supply of wealth management products and has lowered the threshold for farmer households to participate in financial management. After adopting digital payments, farmers can choose digital financial products. The convenience of mobile phones and the Internet has increased farmers' demand for financial products. The supply–demand curve of the financial market has shifted to the right from $f_1$ to $f_2$, and farmers' demand for financial products has increased by $e_1$, which is called the "supply effect" of digital payment on investment. On the other hand, the use of digital payments has changed the way farmers purchase financial products. Farmer households can obtain and process relevant information about investments in a timely manner, which reduces the cost of participating in the financial market and has flattened the supply and demand curve of the financial market. The supply–demand curve of the financial market has changed from $f_2$ to $f_3$. At this point, the demand of farmer households for financial

products has increased by $e_2$, which is called the "information effect". It can be seen that the equilibrium point has finally moved from $Q_1$ to $Q_2$, which means that the demand for investment of farmer households has ultimately increased by $Q_2$–$Q_1$.

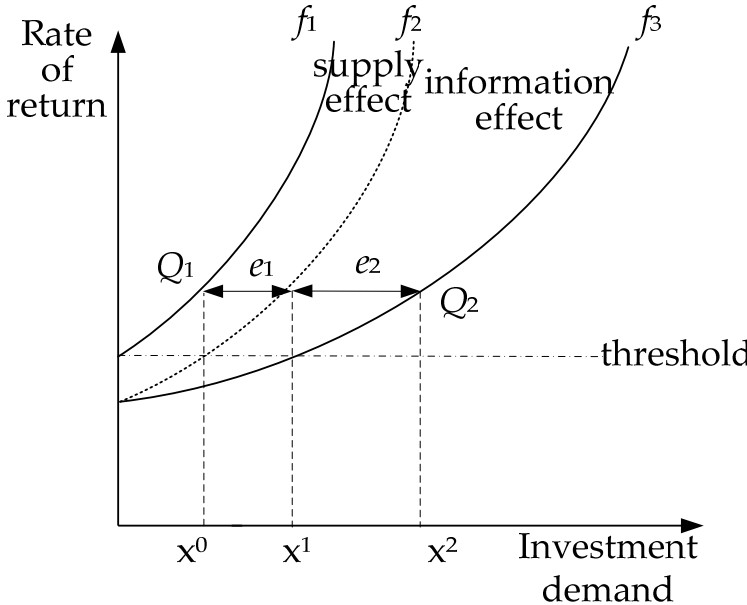

**Figure 1.** The mechanism of the impact of digital payment on the investments of farmer households.

The internal mechanism of how the use of digital payment affects the bank loan behavior of farmer households is shown in Figure 2. Unlike investment, the impact of digital payment on bank loans is first realized by changing the method of information transmission. On the one hand, digital payment alleviates the information asymmetry in the financial market, thereby reducing the resulting transaction costs [33]. It improves the financial knowledge of farmers and reduces implicit financial exclusion due to lack of financial knowledge. On the other hand, digital technology increases the accessibility of financial institutions by reducing the operating costs of banks. The effects of these two aspects have made it possible to flatten the supply–demand curve without increasing the supply in the financial market. The bank loan supply–demand curve has changed from $l_1$ to $l_2$, and the bank loan demand of the farmer households has increased by $e_3$. Digital technology pushes the bank loan demand of farmer households to the back end of the long tail, and the inclusion effect is more obvious for disadvantaged groups closer to the bottom of the demand market. The inclusive mechanism by which digital payment has improved information efficiency and reduced financial costs is called the "information effect". Traditional financial institutions can use digital technology to create more inclusive credit products through low-cost operations, increasing the supply of loans to a certain extent. Farmers who use digital payments can use these new supplies to meet their borrowing needs, making the supply–demand curve of financial market shift right from $l_2$ to $l_3$, and the demand for bank loans increases by $e_4$. The mechanism of the inclusive effect due to increased supply is called the "supply effect". The use of digital payment eventually makes the equilibrium point move from $P_1$ to $P_2$, that is, the demand for bank loans of farmer households eventually increases by $P_2$–$P_1$.

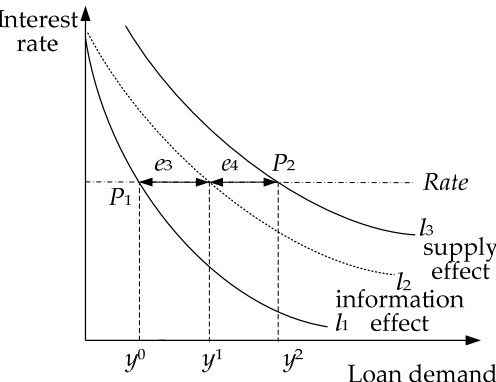

**Figure 2.** Mechanism of the impact of digital payment on the bank loans of farmer households.

## 3. Research Background: Digital Finance Usage and Financial Exclusion

### 3.1. Data Description

The data were drawn from the "Rural Inclusive Finance Survey" conducted by China Agricultural University from 2017 to 2019. The survey adopted a multistage random sampling approach. We randomly chose one province each from the eastern, central, and western regions of China, representing advanced, middle, and low development levels in China, respectively. In 2017, Shandong Province, Henan Province, and Guangxi Province were selected, and in 2018 and 2019, Shandong Province, Henan Province, and Guizhou Province were selected. In each province, we ranked the counties based on the per capita gross domestic product (GDP) from the highest to the lowest, and equally divided the counties into high-, middle-, and low-level groups. The top, middle, and bottom counties were chosen to capture an equal representation of various levels of per capita GDP by population. Hence, using the same method, three townships were selected according to the level of per capita GDP in each county. In each township, we randomly select two natural villages, in which the population is around 30–50 households, and surveyed all villagers. In total, 5800 questionnaires were collected, including 2029 from 2017, 1975 from 2018, and 1733 from 2019. In order to improve the representativeness of the sample, this study excluded the sample of farmers aged younger than 16 or older than 60. The remaining questionnaires numbered 4445, including 1622 from 2017, 1504 from 2018, and 1319 from 2019. After data cleaning, 4178 questionnaires were used in total, including 1506 from 2017, 1420 from 2018, and 1252 from 2019.

### 3.2. Current Status of Digital Finance and Digital Payment Usage

Table 1 provides a description of the current situation of farmer households using digital finance and digital payment. From Table 1, we can see that the proportion of rural households with smartphones was 56.44%, 65.99%, and 74.76%, respectively, for 2017, 2018, and 2019. The increasing number of people using smartphones indicates that more farmer households have the conditions for the adoption of digital finance. In terms of digital financial behavior, 30.88% of farmers used digital payments in 2017, and this proportion rose to 61.74% in 2019, indicating that digital payments are generally accepted by farmer households. Compared with 2017, the proportion of investments and borrowing using mobile technology in 2019 has increased but has remained below 5%, indicating that only a few farmer households use digital technology to engage in financial activities other than payments. Through further analysis of digital payment, it was found that the current payment methods used by farmers in China are mainly WeChat Pay (WeChat 6.5.3~7.0.10, Shenzhen city Tencent computer system Co., Ltd., Shenzhen, China) and Alipay (Alipay 10.0.0.122205~10.1.81.7020, Alipay.com Co., Ltd, Hangzhou, China). In 2019, these two payment methods were used by 61.34% and 41.93% of the investigated farmers, respectively. Mobile banking and electronic wallets (such as Baidu Wallet (Baidu Wallet 3.2.0~3.6.3, Beijing Baidu Netcom Science and Technology Co., Ltd., Beijing, China), JD

Wallet (JD Wallet 5.1.8~6.5.2, Online Banking (Beijing) Technology Co., Ltd., Beijing, China) and Yipay (Yipay 6.0.3~9.10.1, Tianyi E-commerce Co., Ltd., Beijing, China)) are being used relatively more often, accounting for 22.76% and 7.03%, respectively, in 2019.

**Table 1.** The use of digital finance and digital payment by rural households.

| Digital Finance | 2017 | 2018 | 2019 | Digital Payment | 2017 | 2018 | 2019 |
|---|---|---|---|---|---|---|---|
| Smartphone | 56.44% | 65.99% | 74.76% | Mobile banking | 10.69% | 14.30% | 22.76% |
| Digital payment | 30.88% | 45.28% | 61.74% | WeChat Pay | 27.16% | 43.73% | 61.34% |
| Investments | 0.27% | 0.21% | 1.92% | Alipay | 21.05% | 32.04% | 41.93% |
| Borrowing | 0.66% | 0.42% | 4.47% | Electronic wallet | 0.73% | 0.92% | 7.03% |
| | | | | Other | 0.20% | 0.28% | 5.35% |

*3.3. Analysis of Farmer Households Facing Financial Exclusion*

The concept of financial exclusion was first proposed by Leyshon and Thrift [34]. It was originally aimed at the problem of banks closing branches in remote areas and affecting people's access to financial services. Subsequently, discussions of financial exclusion have mainly revolved around the financial availability of disadvantaged groups, and has been extended to broader financial discrimination, such as availability exclusion and accessibility exclusion [35], active exclusion, and passive exclusion [5]. Drawing lessons from the classification of financial exclusion in Kempson and Whyley [35], combined with the current status, the investment exclusion faced by farmer households is divided into knowledge exclusion and conditional exclusion, and bank loan exclusion is divided into knowledge exclusion, price exclusion, and conditional exclusion. The specific descriptions of the types of financial exclusion are shown in Table 2.

**Table 2.** Types, performance, and reasons for farmer households' financial exclusion.

| Types | Specific Types | Behavior and Reasons |
|---|---|---|
| Investment exclusion | Knowledge exclusion | No relevant knowledge, not knowing how to open an account, not knowing where to open an account |
| | Conditional exclusion | The account opening procedure is cumbersome, the security company is too far away, and the funds are limited |
| Bank loan exclusion | Knowledge exclusion | Failure to apply for a loan from the bank because of not knowing how to apply for a loan. |
| | Price exclusion | Not applying for loans from banks due to restrictions on explicit costs such as an excessively long application process, high interest rates and other hidden costs that are difficult to quantify |
| | Conditional exclusion | Not applying for or receiving loans from the bank due to restrictions such as a lack of collateral, being unable to find a guarantor, no social relationship with the bank staff or fear of repayments because of low income. |

From Table 3, we can see that 3609 households were excluded from the investment market, accounting for 86.38% of the surveyed farmer households. This shows that most households in rural areas have obstacles to asset allocation and investment activities. The exclusion from the investment market is mainly manifested as knowledge exclusion, accounting for 72.14% of the surveyed households. In addition, 21.47% of rural households still face conditional exclusion in the investment market. Compared with the 2017 survey data, the proportion of households facing investment exclusion has declined in 2018 and 2019.

**Table 3.** The status quo of farmers facing investment exclusion.

| Investment Exclusion | 2017 | | 2018 | | 2019 | | Number of Households | Percentage |
|---|---|---|---|---|---|---|---|---|
| | Number | Percentage | Number | Percentage | Number | Percentage | | |
| All exclusion | 1342 | 89.11% | 1203 | 84.72% | 1064 | 84.98% | 3609 | 86.38% |
| Knowledge exclusion | 1114 | 73.97% | 1007 | 70.92% | 893 | 71.33% | 3014 | 72.14% |
| Conditional exclusion | 348 | 23.11% | 300 | 21.13% | 249 | 19.89% | 897 | 21.47% |

From Table 4, we can see that there were 832 rural households excluded from the bank loans market, accounting for 19.91% of the investigated households. Bank loan exclusion mainly manifested as conditional exclusion and knowledge exclusion, accounting for 10.53% and 6.99% of surveyed farmer households, respectively. In addition, 4.04% of farmer households faced price exclusion. Compared with the 2017 survey data, the proportion of farmer households facing bank loan exclusion declined in 2018 and 2019. On the whole, exclusion from the investment market was more serious than exclusion from the bank loan market, and the lack of corresponding financial knowledge was the main reason why farmers faced exclusion.

**Table 4.** The current situation of rural households facing bank loan exclusion.

| Bank Loan Exclusion | 2017 | | 2018 | | 2019 | | Number of Households | Percentage |
|---|---|---|---|---|---|---|---|---|
| | Number | Percentage | Number | Percentage | Number | Percentage | | |
| All exclusion | 407 | 27.03% | 221 | 15.56% | 204 | 16.29% | 832 | 19.91% |
| Knowledge exclusion | 148 | 9.83% | 79 | 5.56% | 65 | 5.19% | 292 | 6.99% |
| Price exclusion | 71 | 4.71% | 54 | 3.80% | 44 | 3.51% | 169 | 4.04% |
| Conditional exclusion | 225 | 14.94% | 110 | 7.75% | 105 | 8.39% | 440 | 10.53% |

## 4. Measurement of Farmer Households' Financial Availability Index

### 4.1. Construction of the Index of Financial Availability

In order to comprehensively portray the financial status of rural households, all of these dimensions needed to be examined, along with the causes of all of the barriers—price and non-price—to financial inclusion [5]. This study measured the financial availability of farmer households in terms of three dimensions: investment availability, bank lending availability, and availability of private finance. The investment availability indicators were measured by five specific indicators, including investment knowledge, number of deposit accounts, bank deposit types, investment types and investment exclusion. Among these, investment exclusion is a negative index, and the rest are positive indicators. The bank loan availability indicators were measured by five specific indicators: bank loan knowledge, bank credit rating, bank credit line, bank loan status, and bank loan exclusion. Bank loan exclusion is a negative indicator, and the rest are positive indicators. The private finance availability index was measured by three specific indicators: private borrowing capacity, borrowing situation, and lending situation, all of which are positive indicators. Specific descriptions of each indicator are shown in Table 5. Except for private borrowing situation, the availability indicators of farmers who use digital payments were significantly better than those who do not use digital payments.

**Table 5.** Index of rural households' financial availability.

| Dimensions | Variables | Variable Definitions | Dig. Pay. Mean | Non-Dig. Pay. Mean | Difference |
|---|---|---|---|---|---|
| Investment availability IA | Investment knowledge | The number of correct answers to the four financial management questions | 0.427 | 1.007 | −0.580 *** |
| | Number of deposit accounts | Number of bank deposit accounts | 1.172 | 2.011 | −0.838 *** |
| | Bank deposit types | How many of the following types of deposits the farmer has: fixed deposits, demand deposits and bank wealth management products | 0.827 | 1.063 | −0.236 *** |
| | Investment product types | How many of the following types of financial products there are: bonds, funds, trust and asset management products, non-RMB assets, gold, derivatives, commercial insurance, stocks, and Internet crowdfunding products | 0.013 | 0.1 | −0.087 *** |
| | Investment exclusion | Does investment face knowledge exclusion or conditional exclusion (1 = yes; 0 = no) | 0.915 | 0.801 | 0.114 *** |
| Bank loan availability BLA | Bank loan knowledge | The level of understanding of the conditions and procedures of bank loans: 1. not at all; 2. not much understanding; 3. understanding; 4. relatively good understanding; 5. very good understanding | 2.04 | 2.74 | −0.700 *** |
| | Bank credit rating | Has the farmer received a bank credit rating? (1 = yes; 0 = no) | 0.229 | 0.464 | −0.235 *** |
| | Bank credit line | What is the credit line? (10,000 yuan) | 1.826 | 6.274 | −4.448 *** |
| | Bank loan situation | Has the farmer obtained a loan from a bank or a rural credit cooperative (1 = yes; 0 = no) | 0.188 | 0.32 | −0.132 *** |
| | Bank loan exclusion | Does the farmer face bank loan knowledge exclusion, price exclusion or condition exclusion? (1 = yes; 0 = no) | 0.251 | 0.136 | 0.116 *** |
| Private finance availability PFA | Private borrowing capability | If someone in the family is sick and needs funds urgently, how many people can the farmer ask for help? | 6.558 | 7.804 | −1.246 *** |
| | Private borrowing situation | Has the farmer borrowed money from other people, cooperatives or institutions? (1 = yes; 0 = no) | 0.211 | 0.229 | −0.018 |
| | Private lending situation | Has the farmer lent money to others ("others" refers to people or institutions other than family members)? (1 = yes; 0 = no) | 0.112 | 0.255 | −0.143 *** |

Notes: (1) The specific questions about investment were as follows: (I) Assuming that you have deposited 100 yuan in the bank, the bank interest rate is 2% and the deposit has been for 5 years, how much money is in your account after 5 years: (a) equal to 110; (b) greater than 110; (c) less than 110; (d) not sure. (II) If the bank interest rate is 10% and the inflation rate is 12%, can we buy more or less than a year ago if we take out the money in one year: (a) more things; (b) the same; (c) fewer things; (d) not sure. (III) If the exchange rate of USD to RMB is 1:6, how much USD is equivalent to 600 RMB? (a) 100; (b) 3600; (c) 60; (d) not sure. (IV) Which is a fixed-income financial product: (a) stocks; (b) bonds; (c) funds; (d) not sure. (2) Credit ratings and credit lines are generally operated and determined by rural commercial banks. This is a policy implemented by rural commercial banks in rural areas to evaluate the credit of rural households and determine the amount of loans available according to their household conditions. (3) ***: Significant level at 1%.

### 4.2. Calculated Results of the Financial Availability Index for Farmer Households

Using the survey data from 2017 to 2019, the results of the farmer households' financial availability index and the availability index of each dimension calculated according to the Cov-AHP weighting method [36,37] and the Euclidean distance method are shown in Table 6 (The calculation process is detailed in the Appendix A). The average value of the farmer households' financial availability index was 0.192, and the average value of the investment, bank loan, and private finance availability indices were 0.116, 0.207, and 0.124, respectively. From the perspective of changes over time, except for the decline in

the private finance availability index, the investment availability index, the bank loan availability index, and the financial availability index have all increased.

**Table 6.** Results of the rural households' financial availability index.

| Dimensions | 2017 | | 2018 | | 2019 | | All Samples | |
|---|---|---|---|---|---|---|---|---|
| | Mean | Variance | Mean | Variance | Mean | Variance | Mean | Variance |
| IA | 0.111 | 0.082 | 0.122 | 0.085 | 0.115 | 0.078 | 0.116 | 0.082 |
| BLA | 0.168 | 0.145 | 0.230 | 0.161 | 0.227 | 0.158 | 0.207 | 0.157 |
| PFA | 0.127 | 0.113 | 0.136 | 0.119 | 0.105 | 0.093 | 0.124 | 0.110 |
| Composite index: | | | | | | | | |
| IFA | 0.178 | 0.101 | 0.207 | 0.108 | 0.190 | 0.098 | 0.192 | 0.103 |

## 5. Empirical Framework

### 5.1. Variables

This study took farmer households' financial availability index as the dependent variable and the adoption of digital payment as the independent variable. Financial availability is represented by the financial availability index and the three dimensions of the availability index of rural households.

With reference to the relevant literature on financial availability [22,38], we controlled for variables such as individual, household, social relations, and wealth variables that could impact farmer households' financial availability, as well as the area of the county where the farmer is located and the year of the survey. The individual variables included gender, age, and years of education. The household variables included household size, college students, household income, and the proportion of agricultural income. The social relations refer to whether someone in the household is engaged in non-agricultural industries. The wealth variable is embodied in two aspects: whether the farmer has a commercial house or whether she/he has a car for her/his own use. In order to eliminate the influence of individual values, household income was reduced at the 1% level. In order to overcome the problem of excessive data differences and heteroscedasticity, natural logarithmic transformation of this variable was carried out in the process of empirical analysis. The definition and descriptive statistics of the variables are shown in Table 7.

**Table 7.** Definition and descriptive statistics of the control variables.

| Variables | Definitions | Mean | Std. Dev |
|---|---|---|---|
| Gender | Gender of the respondent (male = 1; female = 0) | 0.544 | 0.498 |
| Age | Age of the respondent | 46.51 | 9.373 |
| Education | Respondents' years of formal education | 7.904 | 3.265 |
| Family | The number of family members | 4.539 | 1.634 |
| Work-Prop | Proportion of the working population in the family | 0.652 | 0.240 |
| College | Are there any university students in the home? (yes = 1; no = 0) | 0.230 | 0.421 |
| Income | The logarithm of the family's actual income in the previous year, including agricultural income and other income (yuan) | 10.38 | 1.326 |
| Agri-Ratio | The ratio of household income from planting and breeding in the previous year to annual household income | 0.314 | 0.395 |
| Soci-Relations | Whether someone in the household is engaged in non-agricultural industries (yes = 1; no = 0) | 0.449 | 0.497 |

**Table 7.** *Cont.*

| Variables | Definitions | Mean | Std. Dev |
|---|---|---|---|
| Comm-House | Whether the respondent owns a commercial house in the city (yes = 1; no = 0) | 0.128 | 0.334 |
| Car | Whether the respondent owns a car (yes = 1; no = 0) | 0.403 | 0.491 |

Notes: (1) The non-agricultural industries include long-term employment in local or non-local enterprises; household entrepreneurship (business in local or non-local areas); teachers or doctors; working in a cadre in the county or village. (2) The workable population refers to healthy family members aged 18–60.

*5.2. Estimation Technique*

This study focused on the impact of the use of digital payments on the financial availability index of rural households, including the impact on the investment availability, bank loan availability, and private finance availability of farmer households. For this purpose, the following benchmark regression model was constructed:

$$IFA_i = \beta digpayment_i + \gamma controls + \mu_i \tag{1}$$

Consistent with Sarma and Pais [39], we took the logarithm of the rural household financial availability index to expand the distribution interval, and constructed the logarithmic regression model as follows:

$$ln\left(\frac{IFA_i}{1 - IFA_i}\right) = \beta digpayment_i + \gamma controls + \mu_i \tag{2}$$

We used the ordinary least squares method to estimate the benchmark regression and logarithmic regression.

## 6. Estimated Results and Discussion

Because the data used in this study are mixed panel data, the standard errors of all regressions were estimated using farmer-level clustering of robust standard errors, that is, the same farmer household is allowed to have a correlation but different farmer households are not.

*6.1. The Adoption of Digital Payment and Financial Availability*

As a first step toward measuring the effect of digital payment adoption on rural households' financial availability index, we controlled for individual, household and social relations, and wealth variables. The results are reported in Column (1) of Table 8 and show that the coefficient of the impact of digital payment adoption on farmer households' financial availability is 0.043. In Specification 2, we further controlled for the fixed effects of year and location. The coefficient of the effect of digital payment adoption on the financial availability index becomes 0.045 and remains significant at the 1% level. In Specification 3, using the logarithmic form of the rural household financial availability index, the results show that the adoption of digital payment has increased the financial availability index of farmers by 32.2%. In Specification 4, we focused on an analysis of the impact of digital payment usage frequency on the financial availability of farmers. As shown in Column (4) of Table 8, the coefficients show an upward trend as the frequency of use of digital payments increases. This finding shows that the adoption of digital payment has improved the financial availability of farmer households, and the more frequently farmers use digital payments, the higher their financial availability.

**Table 8.** Digital payment adoption and financial availability (estimated by OLS models).

| | (1) | (2) | (3) | (4) |
|---|---|---|---|---|
| | IFA | IFA | Ln IFA | IFA |
| Digital payment | 0.043 *** | 0.045 *** | 0.322 *** | |
| Frequency: | (0.003) | (0.003) | (0.030) | |
| Less than once a week | | | | 0.031 *** |
| | | | | (0.009) |
| Once or twice a week | | | | 0.029 *** |
| | | | | (0.010) |
| Three to five times a week | | | | 0.032 *** |
| | | | | (0.009) |
| Use every day | | | | 0.057 *** |
| | | | | (0.007) |
| Control variables | Yes | Yes | Yes | Yes |
| Year dummies | No | Yes | Yes | No |
| Location dummies | No | Yes | Yes | Yes |
| R-squared | 0.289 | 0.340 | 0.233 | 0.340 |
| Observations | 4178 | 4178 | 4178 | 1252 |

Notes: (1) ***: Significant level at 1%. (2) Family-level clustered robust standard errors are given in parentheses in Columns (1) to (3). (3) Columns (1) to (3) present the results based on the full sample, while the sample in Column (4) is limited to survey data from 2019.

### 6.2. Digital Payment Adoption and the Availability of Investments, Bank Loans, and Private Finance

The results of Columns (1), (3), and (5) in Table 9 show that the coefficients of the influence of digital payment adoption on the availability of investment, bank loans, and private finance are 0.037, 0.051, and 0.019, respectively. Obviously, the adoption of digital payments has a greater impact on the availability of investments and bank loans. The regression results of Columns (2), (4), and (6) once again show that the higher the frequency of using digital payments, the greater the coefficient of influence on the financial availability of rural households. If farmers use digital payments less frequently than once a week, the use of digital payments will not have a significant impact on the investment and private finance availability of farmer households, but it will still have a significant impact on the availability of bank loans. This finding shows that the adoption of digital payment has improved the availability of investment, bank loans, and private finance for farmers; the more frequent the use of digital payment, the more obvious the positive effect of digital payment; farmers who use digital payment every day have higher financial management, credit, and private availability.

**Table 9.** Digital payment adoption and the investment, bank loan, and private finance availability indices (estimated by OLS models).

| | (1) | (2) | (3) | (4) | (5) | (6) |
|---|---|---|---|---|---|---|
| | Investment Availability | | Bank Loan Availability | | Private Finance Availability | |
| Digital payment | 0.037 *** | | 0.051 *** | | 0.019 *** | |
| Frequency: | (0.003) | | (0.005) | | (0.004) | |
| Less than once a week | | 0.012 | | 0.049 *** | | 0.013 |
| | | (0.008) | | (0.015) | | (0.010) |
| Once or twice a week | | 0.030 *** | | 0.026 * | | 0.005 |
| | | (0.008) | | (0.016) | | (0.011) |
| Three to five times a week | | 0.039 *** | | 0.024 * | | 0.008 |
| | | (0.007) | | (0.014) | | (0.009) |

**Table 9.** *Cont.*

|  | (1) | (2) | (3) | (4) | (5) | (6) |
|---|---|---|---|---|---|---|
|  | Investment Availability | | Bank Loan Availability | | Private Finance Availability | |
| Use every day |  | 0.041 *** |  | 0.057 *** |  | 0.033 *** |
|  |  | (0.006) |  | (0.011) |  | (0.007) |
| Control variables | Yes | Yes | Yes | Yes | Yes | Yes |
| Year dummies | Yes | No | Yes | No | Yes | No |
| Location dummies | Yes | Yes | Yes | Yes | Yes | Yes |
| R-squared | 0.295 | 0.295 | 0.330 | 0.330 | 0.104 | 0.104 |
| Observations | 4178 | 1252 | 4178 | 1252 | 4178 | 1252 |

Note: * and *** indicates significance at the 1% level.

### 6.3. Digital Payment Adoption, Information Effects and Financial Availability

The theoretical analysis explains the internal mechanism, namely the "supply effect" and the "information effect", of how digital finance affects the financial availability of farmers. The "supply effect" is a direct effect, and it is reflected by the provision of more financial channels. The status quo of the use of digital finance by farmers indicates that the supply effect has little impact on farmers at the current stage. The information effect is an indirect effect, which is mainly achieved through changes made by farmers and banking institutions. This study believes that the impact of the information effect on farmers is multifaceted, and can be reflected by the promotion of and changes in farmers' information acquisition and information attention. This study introduced two proxy variables, "information channel" and "information attention", to test the internal mechanism of how digital payments affect the financial availability of farmers. The information channel variable was obtained by asking farmers how many channels they used to obtain information (including: newspapers or magazines; television; radio; Internet; mobile phone text messages; relatives, friends and colleagues; and other). The information attention variable was obtained by asking farmers how many different aspects of information they pay attention to (including: subsidy policies, agricultural material prices, meteorological information, employment information, agricultural insurance policies, pensions, rural cultural life, education, healthcare, and other). First, we analyzed whether the adoption of digital payment has affected farmers' information channels and information attention. We examined the impact of digital payment adoption on the information channel and information attention using the following regression:

$$\Pr(Information\ effect_i^* = Information\ effect_i) = G(\beta digpayment_i + \lambda controls + \varepsilon) \quad (3)$$

where information effect$^*$ is an unobservable variable, but information effect is an ordered variable reflected by information channels and information attention. Because information channels and information attention are both ordered variables, the ordered probit method was used to calculate the coefficients of influence. The results of Equation (3) are shown in Table 10. These results suggest that the adoption of digital payment has a positive impact on information channels and information attention. This finding suggests that the adoption of digital payment has helped farmers to expand the channel of information acquisition and pay attention to a broader range of information.

**Table 10.** Digital payment adoption and information effects (estimated by ordered probit models).

|  | (1) | (2) | (3) | (4) |
|---|---|---|---|---|
|  | **Information Channel** | | **Information Attention** | |
| Digital payment | 0.301 *** | 0.310 *** | 0.128 *** | 0.160 *** |
|  | (0.041) | (0.042) | (0.040) | (0.041) |
| Control variables | Yes | Yes | Yes | Yes |
| Year dummies | No | Yes | No | Yes |
| Location dummies | No | Yes | No | Yes |
| Pseudo-$R^2$ | 0.030 | 0.043 | 0.009 | 0.028 |
| Observations | 4178 | 4178 | 4178 | 4178 |

Note: *** indicates significance at the 1% level.

The results in Table 10 demonstrate that the use of digital payment has an information effect and can improve the information acquisition ability of farmers. We next examined whether the adoption of digital payment can affect the financial availability of farmers through information effects by running the following regression:

$$IFA_i = \beta digpayment_i + \gamma Information\ effect_i + \lambda controls_i + \varepsilon \qquad (4)$$

Columns (1) and (3) of Table 11 indicate that having more information acquisition channels and information attention can improve the financial availability of farmers. Columns (2) and (4) report the results of Equation (4). Specifically, the results in column (2) show that the adoption of digital payment can increase the farmers' financial availability by increasing the channels by which farmers acquire information. The results of column (4) show that the adoption of digital payment can improve the financial availability of farmers by prompting farmers to pay more attention to more information. This analysis verified the mechanism of influence of digital payments to improve the financial availability of farmers through the information effect.

**Table 11.** Digital payment adoption, information effects and financial availability (estimated by OLS models).

|  | (1) | (2) | (3) | (4) |
|---|---|---|---|---|
|  | **Financial Availability** | | **Financial Availability** | |
| Digital payment |  | 0.043 *** |  | 0.044 *** |
|  |  | (0.003) |  | (0.003) |
| information channel | 0.010 *** | 0.008 *** |  |  |
|  | (0.001) | (0.001) |  |  |
| information attention |  |  | 0.005 *** | 0.004 *** |
|  |  |  | (0.001) | (0.001) |
| Control variables | Yes | Yes | Yes | Yes |
| Year dummies | Yes | Yes | Yes | Yes |
| Location dummies | Yes | Yes | Yes | Yes |
| R-squared | 0.319 | 0.345 | 0.317 | 0.344 |
| Observations | 4178 | 4178 | 4178 | 4178 |

Notes: *** indicates significance at the 1% level.

*6.4. Robustness Check*

6.4.1. Unobservable Characteristics and Model Misspecification

Our previous estimations were based on a key identification assumption that the dependent variable has a linear relationship with the covariates. However, our previous estimators may be biased if this assumption does not hold. In this section, we used the endogenous switching regression models (ESR) proposed by Lokshin and Sajaia [40] to minimize the problem of model misspecification. Compared with the PSM model, the

ESR model can simultaneously estimate two sets of equations to evaluate the impact of digital payment and non-use of digital payment on the financial availability of farmers. Through the full-information maximum likelihood estimation, the unobservable biases are incorporated into the selection model to correct the selection bias, thus helping to avoid the problem of missing effective information. The model included the digital payment usage selection equation and the equation for determining farmer households' financial availability. The specific calculation process was as follows: first, we used the binary choice model to estimate the probability of farmers using digital payment; second, we estimated the equation for determining the farmers' financial availability index in the two cases of using digital finance and not using digital payments; finally, the average treatment effect on the treated (ATT) and average treatment effect on the untreated (ATU) were estimated according to the estimation results. The estimated ATTs and ATUs, as reported in Table 12, are consistent with our previous results.

**Table 12.** Digital payment adoption and financial availability (estimated by ESR models).

|  | Financial Availability | Investment Availability | Bank Loan Availability | Private Finance Availability |
|---|---|---|---|---|
| ATT | 0.075 *** | 0.070 *** | 0.076 *** | 0.025 *** |
|  | (0.001) | (0.000) | (0.001) | (0.001) |
| ATU | 0.058 *** | 0.041 *** | 0.107 *** | −0.000 |
|  | (0.000) | (0.000) | (0.001) | (0.000) |

Notes: *** indicates significance at the 1% level.

Digital payment adoption has an impact coefficient (ATT) of 0.075 on the financial availability of farmers who use digital payments, and an impact coefficient (ATU) of 0.058 for farmers who do not use digital payments. That is, compared with the counterfactual situation of digital payment farmers, the financial availability index is, on average, higher than 0.075. At the same time, the financial availability index of farmers who do not use digital payment is 0.058 lower than the counterfactual situation on average. Specifically, the ATT and ATU of digital payment on the investment availability of farmers are 0.070 and 0.041, respectively. The ATT and ATU of digital payment on the bank loan availability of farmers are 0.076 and 0.107, respectively. The ATT of digital payment on the private finance availability of farmers is 0.076, and the ATU is not significant. Similar to the previous results, the adoption of digital payment has the greatest coefficient for the impact on bank loan availability, followed by the coefficient of the impact on investment availability, and has the smallest coefficient for the impact on private finance.

6.4.2. Endogenous Explanation and Resolution

However, there may be endogenous problems with the farmers' financial availability index and the use of digital payments. First, there may be variables that are difficult to measure that can affect both the level of financial availability of farmers and the use of digital payments, leading to endogenous problems caused by missing variables. Second, farmers may adopt digital payment methods in order to enjoy financial services more conveniently, leading to endogenous problems due to reverse causality. Finally, the survey process may have led to measurement errors in the variables, resulting in endogenous problems. The endogenous problems will lead to endogenous biases, and the biases will not disappear gradually. This study solved the problem of endogeneity by introducing instrumental variables in two specific ways, as described below.

(1) To account for the fact that smartphones are a necessary condition for farmers to adopt digital payments, and that smartphones only affect farmers' financial availability through digital payments, we used the variable of whether the farmer has a smartphone (yes = 1, no = 0) as an instrument for digital payment adoption. The results based on the two-stage least squares (2SLS) model are reported in Table 13. We can see that, after controlling for the reverse causality of financial availability on the adoption of digital payments, the coefficients of the influence of digital payment adoption on financial availability are still

positive and statistically significant at the 1% level. These results reinforce our previous finding that digital payment adoption has a positive effect on financial availability.

**Table 13.** Digital payment adoption and financial availability (estimated by 2SLS models).

| | (1) | (2) | (3) | (4) |
|---|---|---|---|---|
| | **Financial Availability** | **Investment Availability** | **Bank loans Availability** | **Private Finance Availability** |
| Digital payment | 0.107 *** | 0.085 *** | 0.119 *** | 0.041 *** |
| | (0.010) | (0.008) | (0.016) | (0.012) |
| Control variables | Yes | Yes | Yes | Yes |
| Year dummies | Yes | Yes | Yes | Yes |
| Location dummies | Yes | Yes | Yes | Yes |
| R-squared | 0.285 | 0.242 | 0.302 | 0.099 |
| Observations | 4178 | 4178 | 4178 | 4178 |

Notes: *** indicates significance at the 1% level.

(2) Conley et al. [41] believe that instrumental variables can be approximately exogenous. Considering that the instrumental variable for whether the farmer has a smartphone is not strictly exogenous, additional analysis was performed according to the UCI method (union of confidence intervals) proposed by Conley to test the robustness of the estimated results under the condition of imperfect exogenous instrumental variables. The confidence intervals of the regression coefficients are shown in Figure 3. Except for the availability of private finance, the coefficients of the influence of digital payment adoption on the financial availability and the availability of investment and bank loans are always greater than 0.

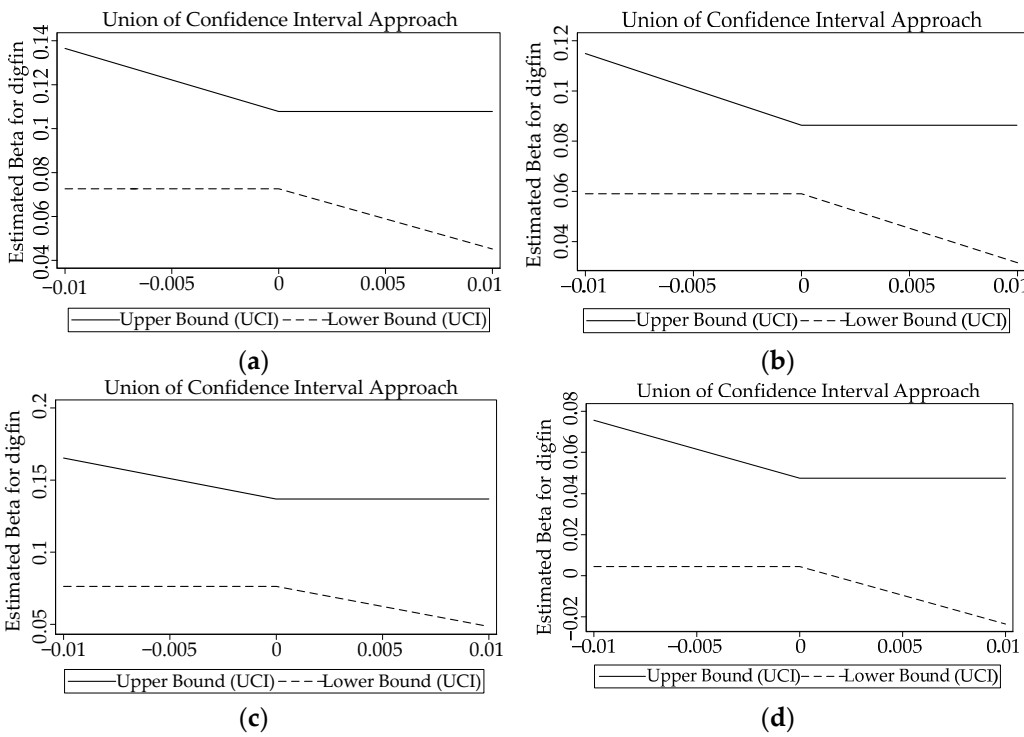

**Figure 3.** Confidence intervals based on the UCI method. (**a**) Financial availability; (**b**) investment availability; (**c**) bank loan availability; (**d**) private finance availability.

## 7. Conclusions and Policy Implications

The financial availability of farmer households is a microscopic manifestation of inclusive rural finance, which directly reflects the situation of financial services enjoyed by farmer households. As the vanguard of digital finance, digital payment is a favorable means for the development of inclusive finance. The analysis of the internal relationship between the adoption of digital payment and the availability of rural households is helpful

for clarifying the internal mechanism of digital finance to achieve inclusive effects. The conclusions of this research provide a reference for using digital finance to improve the financial availability of disadvantaged groups such as farmers, thereby promoting the development of inclusive finance and social equity.

Based on the long tail effect theory, this study analyzed the internal mechanism of how digital payment affects the financial availability of farmer households through the "information effect" and the "supply effect". The level of digital finance usage by farmer households shows that digital payment is the most widely used digital financial function in rural areas. In 2019, 61.74% of farmer households used digital payments. The analysis of financial exclusion shows that 86.38% of farmer households faced investment exclusion, and 19.91% of farmer households faced bank loan exclusion. Among these, investment exclusion was mainly manifested as knowledge exclusion, while bank loan exclusion was mainly manifested as conditional exclusion and knowledge exclusion. The lack of financial knowledge has become an important factor restricting farmer households' enjoyment of financial services. Based on the three dimensions of investment availability, bank loan availability, and private finance availability, an indicator system of financial availability has been constructed and the level of farmer households' financial availability has been measured. The measurement results show that the farmer households' financial availability is still at a low level, the availability of bank loans is relatively high, and the availability of investment is the lowest, but both have an upward trend, and the availability of private lending has gradually declined. The empirical analysis shows that digital payments significantly improved the financial availability of farmer households. The mechanism test confirmed that digital payments can improve the financial availability of farmer households, mainly through information effects. The robustness test confirmed the robustness of these conclusions.

This study's conclusions lead to the following policy implications: first, considering that some farmer households still face financial exclusion, the government should pay attention to the issue of farmers' financial availability, especially to solve the obstacles of farmers' participation in the financial market. Second, both the research results and the reality show that digital technology alone is not enough to improve the financial availability of farmers. The government should provide favorable conditions for the development of digital finance in rural areas, such as strengthening the informatization construction in rural areas. Third, according to the analysis results of the mechanism of digital payment affecting farmer households' financial availability, the government should pay attention to the role of financial knowledge in solving financial exclusion, and expand farmers' information channels, especially online information channels. By improving the financial literacy of farmers, digital payment can play a better role in promoting financial inclusion, and ultimately achieve the purpose of improving farmers' financial availability.

At present, many studies are still focused on analyzing the impact of digital finance or digital payment on rural industries [30] or farmer households' lives [42], while research on digital financial inclusion mechanisms still needs to be supplemented. Based on the construction of the financial availability system for farmers, this study analyzes the impact of digital payment adoption on farmers' financial availability and its mechanism, which enriches the existing research results to a certain extent. However, this study also has shortcomings. The limitations of the study are mainly manifested in two aspects. First, the analysis of endogenous problems is still insufficient. Due to space limitations, the analysis of the heterogeneity of farmers needs to be supplemented. Second, only the data of rural households in China was analyzed, and the difference between rural residents and urban residents regarding the use of digital payment needs to be further analyzed. Further research should attempt to overcome such concerns and can use natural experiments as an exogenous shock to identify the casual effect better, while considering extending the sample to urban residents.

**Author Contributions:** Conceptualization, B.C. and J.R.; data curation, B.C.; methodology, B.C.; validation, B.C.; formal analysis, B.C.; writing—original draft preparation, B.C.; writing—review

and editing, J.R.; supervision, J.R.; funding acquisition, J.R. All authors have read and agreed to the published version of the manuscript.

**Funding:** This study was funded by the Beijing Social Science Foundation of China, grant number 19GLA002, and the new doctoral incubation research project of Beijing Union University, grant number SK20202204.

**Institutional Review Board Statement:** Not applicable.

**Data Availability Statement:** Not applicable.

**Acknowledgments:** The authors thank the anonymous reviewers for their critical and constructive review of the manuscript.

**Conflicts of Interest:** The authors declare no conflict of interest.

## Appendix A. The IFA (Index of Financial Availability) Calculation Process

*Appendix A.1. Using the Cov-AHP Method to Calculate the Weight of Each Indicator*

The basic idea of Cov-AHP is as follows: based on the covariance matrix formed by the element indicators, through transformation, calculation and other means, a judgment matrix reflecting the relative importance of each element that had the characteristics of the analytic hierarchy process is constructed. Then, after mathematical calculation and testing, a certain level of weight relative to the highest level was used to illustrate the relative importance of each quantitative index. The traditional AHP method requires a number of experts to make subjective judgments on the relative importance of the elements of the system based on their own experience and knowledge. The advantage of Cov-AHP is that it overcomes the subjective bias of experts. The steps of Cov-AHP are as follows:

I.　Standardizing each index.

$$\begin{cases} x_{ij} = \frac{A_{ij} - m_{ij}}{M_{ij} - m_{ij}}, & A_{ij} \text{ is a positive indicator} \\ x_{ij} = \frac{M_{ij} - A_{ij}}{M_{ij} - m_{ij}}, & A_{ij} \text{ is a negative indicator} \end{cases} \tag{A1}$$

In Equation (A1), $A_{ij}$ is the actual value of the $j$-th indicator of the $i$-th dimension, $x_{ij}$ is the standardized indicator value ($0 \leq x_{ij} \leq 1$), $m_{ij}$ represents the minimum value of the indicator and $M_{ij}$ is the maximum value of the indicator. Equation (A1) guarantees $0 \leq x_{ij} \leq 1$.

II.　Constructing the covariance matrix. We used the standardized data to calculate the covariance matrix of each indicator to form a covariance matrix. $C_{ij}$ represents the $i$-th row and $j$-th column of the covariance matrix elements, and $C_{ij} = C_{ji}$.

III.　Transforming and calculating the covariance to construct a judgment matrix. First, we divided the covariance $C_{ij}$ of each column by the covariance $C_{ii}$ and transformed it into a relative covariance matrix. Next, the covariance on the diagonal was transformed to 1; then we constructed a judgment matrix $D$ according to $d_{ij} = \frac{c_{ij}^{\xi}}{(c_{ij} \times c_{ji})^{\xi/2}}$, $d_{ji} = \frac{c_{ji}^{\xi}}{(c_{ij} \times c_{ji})^{\xi/2}}$, where $d_{ij}$ represents the $i$-th row and $j$-th column of the judgment matrix, and $\zeta$ is an adjustable parameter. The judgment matrix satisfied the condition that $d_{ij} > 0$; $d_{ii} = 1$; $d_{ij} \times d_{ji} = 1$.

IV.　Calculating the weight of each indicator. The eigenvector corresponding to the largest eigenvalue of the judgment matrix is the weight vector of each index according to the principle of AHP. The square root method was used to solve the eigenvector of the judgment matrix. First, we calculated the product of the elements of each row of the judgment matrix to obtain $M_i$; the calculation formula is $M_i = \coprod_{j=1}^{n} d_{ij}$. Second, we found the $n$-th root of each row of $M_i$; the calculation formula is $W_i^{'} = \sqrt[n]{M_i}$. Third, we normalized $W_i$ to obtain the weight of each indicator, and the specific calculation formula is $W_i = \frac{W_i^{'}}{\sum_{j=1}^{n} W_i^{'}}$.

V. Checking the consistency of the judgment matrix. The numerical consistency of the judgment matrix is an important prerequisite for using AHP to determine the weight of each indicator. For this reason, the consistency of the judgment matrix should be tested. The condition that the judgment matrix meets the consistency is that the maximum characteristic root $\lambda_{\max}$ of the matrix is equal to the number of indicators. Based on this requirement, the degree of deviation of the judgment matrix can be tested by setting the indices *CI* and *CR*.

The calculation steps are:

First, post-multiply the judgment matrix *D* by the weight vector $W = (w_1, w_2, w_3, \dots, w_n)'$ to obtain an n-th order column vector *DW*, then calculate the maximum characteristic root $\lambda_{\max}$ of the judgment matrix D according to the formula $\lambda_{\max} = \frac{1}{n}\sum_{i=1}^{n}\frac{(DW)_i}{w_i}$.

Second, calculate the index *CI* to measure the judgment matrix's deviation from consistency:

$$CI = \frac{\lambda_{\max} - n}{n - 1}$$

Third, calculate the random consistency ratio *CR*: $CR = \frac{CI}{RI}$, where *RI* is the average random consistency index as shown in the Table A1. When *CR* is less than 0.1, it is generally considered that the judgment matrix *D* passes the consistency test; otherwise, the judgment matrix needs to be adjusted, which can be achieved by changing $\zeta$.

**Table A1.** Standards of the average random consistency index.

| n | 1 | 2 | 3 | 4 | 5 | 6 | 7 | 8 | 9 | 10 | 11 | 12 | 13 |
|---|---|---|---|---|---|---|---|---|---|----|----|----|----|
| RI | 0.00 | 0.00 | 0.58 | 0.90 | 1.12 | 1.24 | 1.32 | 1.41 | 1.45 | 1.49 | 1.52 | 1.54 | 1.56 |

Fourth, calculate the weight of each indicator relative to the highest target level (IFA). Given the weight $w_i$ of each availability dimension and the weight $w_{ij}$ of each specific indicator in the availability dimension, the weight of each specific indicator relative to the comprehensive index (IFA) is $w_j = \sum w_i w_{ij}$.

Fifth, run the overall consistency test. If we assume that the consistency index of the i-th availability dimension is $CI_i$, and the random consistency index is $RI_i$, then the consistency ratio of the IFA is $CR_{IFA} = \frac{CI_{IFA}}{RI_{IFA}} = \frac{\sum_i m_i CI_i}{\sum_i m_i RI_i}$. If $CR_{IFA} < 0.10$, it is considered that the ranking of each indicator is reasonable; otherwise, the judgment matrix needs to be adjusted.

The final calculation of the weights of each indicator is shown in Table A2.

**Table A2.** Weight calculation results of the Cov-AHP method.

| Availability Dimension | Weights | Specific Indicator | Weights | |
|---|---|---|---|---|
| | | | Relative to the Availability Dimension | Relative to the Highest Target Level (IFA) |
| Investment availability IFA$_1$ | 0.366 | Investment knowledge | 0.188 | 0.069 |
| | | Number of deposit accounts | 0.214 | 0.078 |
| | | Bank deposit types | 0.195 | 0.071 |
| | | Investment product types | 0.223 | 0.081 |
| | | Investment exclusion | 0.181 | 0.066 |

**Table A2.** *Cont.*

| Availability Dimension | Weights | Specific Indicator | Weights | |
|---|---|---|---|---|
| | | | Relative to the Availability Dimension | Relative to the Highest Target Level (IFA) |
| Bank loan availability IFA$_2$ | 0.307 | Bank loan knowledge | 0.196 | 0.060 |
| | | Bank credit rating | 0.186 | 0.057 |
| | | Bank credit line | 0.238 | 0.073 |
| | | Bank loans situation | 0.189 | 0.058 |
| | | Bank loan exclusion | 0.190 | 0.058 |
| Private finance availability IFA$_3$ | 0.327 | Private borrowing capability | 0.436 | 0.143 |
| | | Private borrowing situation | 0.276 | 0.090 |
| | | Private lending situation | 0.288 | 0.094 |

*Appendix A.2. Calculating the Index of Financial Availability*

According to the Euclidean distance, the financial availability of the i-th dimension $IFA_i$ is computed as follows:

$$IFA_i = 1 - \frac{\sqrt{w_{i1}^2(1-x_{i1})^2 + w_{i2}^2(1-x_{i2})^2 + \cdots w_{in}^2(1-x_{in})^2}}{\sqrt{(w_{i1}^2 + w_{i2}^2 + \cdots w_{in}^2)}} \tag{A2}$$

After weights have been assigned to the dimensions, the final *IFA* is computed as follows:

$$IFA = 1 - \frac{\sqrt{w_1^2(1-IFA_1)^2 + w_2^2(1-IFA_2)^2 + w_3^2(1-IFA_3)^2}}{\sqrt{(w_1^2 + w_2^2 + w_3^2)}} \tag{A3}$$

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
