# Peer review of "Does the Adoption of Digital Payment Improve the Financial Availability of Farmer Households? Evidence from China"

_agriculture, doi:10.3390/agriculture12091468_

Round 1

Reviewer 1 Report

The manuscript entitled “Does the Adoption of Digital Payment Improve the Financial Availability of Farmer Households? Evidence from China” is a very interesting article in which authors have done a good job of writing, analysis, and presentation of results. The authors ask an important question, and the answer to this question (results) has potential policy significance. However, there are some flaws in the article that need to be addressed. Considering this, I have few comments and suggestions for the authors to improve the article’s quality:

1.      Please improve the abstract and add necessary parts so as to present a full snapshot of the study.

2.      The discussion section is missing. Please collate all the findings, discuss in the global and regionsl context, and compare with the latest research.

3.      Please also elaborate on the future research directions after the limitations.

4.      Please provide the policy recommendations based on your research findings.

5.      There are some minor mistakes of grammar and capitalization (see for example Line 54).

Reviewer 2 Report

COMMENTS FOR AUTHORS:

General comment

I congratulate the authors for the effort made.

The study is fascinating, different and current. I appreciate the opportunity to read and comment on the document. I congratulate the authors for the idea, and the work is done. The following suggestions could help improve the document.

0.- Title

a) Authors should consider whether the expressions "Rural finance" and/or “rural financial inclusiveness” could be in the title.

1.- Abstract

a) It would be convenient to highlight why the article is novel and the implications for the corresponding segments or stakeholders.

c) The elements or samples studied must be specified, as well as the methodology used.

d) It could be specified that the Long Tail Effect theory of Anderson has been used.

2.- Introduction

a) Citations or references that support the statements made in the text must be provided. For example, after the following text in lines 27 to 29: “Previous studies on inclusive finance have carried out calculations and analyses from the macro-level of countries and regions. Rural finance, as the weakest link in inclusive finance, has not received the attention it deserves”.

b) An additional effort should be made to add clarity, logical and sequential structure, from the general to the particular. The introduction should make it very clear, at a minimum, the object of study, its importance, necessity and urgency, the state of the issue, the gaps and the objective of the study.

c) All information that does not add value but noise should be removed from the introduction.

3.- Literature review

a) The literature review should focus more on the objective of the study, add more clarity and references, and go a little deeper.

4.- Materials and Methods

a) The methodology used should also be better explained and justified.

5.- Results and discussion

a) The passage from the methods and the sample to the results obtained is not clearly understood. How did the methods lead to those results?

b) The discussion and results should relate exclusively to the objectives of the study.

c) The discussion should be related to the objectives, to the review of the literature and to the hypotheses that the authors could include.

6.- Conclusions

a) Some implications should be included, which could be divided into theoretical, practical and methodological.

7.- References

a) The bibliography and citations in the document should be adapted to the journal's requirements, both in form and content.

Thank you very much.

Reviewer 3 Report

The article deals with a very important topic for scientific discussion, but unfortunately I have to reject this work and recommend resubmission. 

The rejection is due in particular to the following reasons: 

1) Submitting an article is the final process of a scholarly work and finding superficiality is not acceptable. You cannot hand in work where you start sentences with lowercase letters (line 54) or where you do not standardise the representation of thousands (line 184, either put a comma or do not). It is not a question of details or only of form, but also of substance. 

2) The introduction and conclusion of the article are not well articulated. It is not very clear what the contribution is and what the novelty is, furthermore, there is no real discussion of the results, and in the conclusion the literature is never referred to. The central part of the paper is a stylistic exercise of statistical models, which if not contextualised in conclusions remains an end in itself. 

For these reasons, I ask the authors to make a greater effort before resubmitting the article

Reviewer 4 Report

The article presents an interesting topic, focused on the connection between digital payments and the financial availability of farm households. Chapters Introduction and Literature adequately explain the problem and theoretical background. Furthermore, the authors present the results of their research based on the Rural Inclusive Finance Survey, for which they use appropriate statistical methods. The chapter Conclusion contains a discussion of the problem, and the authors also mention the limitations of the presented study.

Round 2

Reviewer 3 Report

The text compared to the first version has improved considerably. 

The form problems have all been resolved and the introductory and conclusion sections have also seen a substantial improvement. 

the article may need some minor improvements in the background part. I recommend reading these two articles to enrich the literature and how to systematise the theory part:

Giulio Fusco, Yari Vecchio, Donatella Porrini, Felice Adinolfi, 2021, Improving the economic sustainability of Italian Farmer: an Empirical Analysis of decision-making models for insurance adoption. New Medit: Mediterranean Journal of Economics, Agriculture and Environment= Revue Méditerranéenne dʹEconomie Agriculture et Environment, 20, 3

Frascarelli, Angelo, Simone Del Sarto, and Giada Mastandrea. 2021. A New Tool for Covering Risk in Agriculture: The Revenue Insurance Policy. Risks 9: 131.
